# 70 Years of LD-Steelmaking—Quo Vadis?

**Jürgen Cappel** [1], **Frank Ahrenhold** [2], **Martin W. Egger** [3], **Herbert Hiebler** [4] **and Johannes Schenk** [4,5,*]

1. CSC GmbH, 81673 München, Germany; juergen.cappel@cappel-consult.com
2. ThyssenKrupp Steel Europe AG, 47166 Duisburg, Germany; frank.ahrenhold@thyssenkrupp.com
3. Voestalpine Stahl GmbH, 4020 Linz, Austria; martin.egger@voestalpine.com
4. Ferrous Metallurgy, Montanuniversität Leoben, 8700 Leoben, Austria; heribert.hiebler@unileoben.ac.at
5. K1-Met GmbH, 4020 Linz, Austria
* Correspondence: johannes.schenk@unileoben.ac.at; Tel.: +43-664-808982200

**Abstract:** Basic Oxygen Furnace (BOF) steelmaking is, worldwide, the most frequently applied process. According to the world steel organization statistical report, 2021, it saw a total production share of 73.2%, or 1371.2 million tons per year of the world steel production in 2020. The rest is produced in Electric Arc Furnace (EAF)-based steel mills (26.3%), and only a very few open-hearth and induction furnace-based steel mills. The BOF technology remains the leading technology applied based on its undoubted advantages in productivity and liquid steel composition control. The BOF technology started as the LD process 70 years ago, with the first heat applied in November 1952 in a steel mill in Linz, Austria. The name LD was formed from the first letters of the two sites with the first industrial scale plants, Linz and Donawitz, both in Austria. The history and development of the process have been honored in multiple anniversary publications over the last few decades. Nevertheless, the focus of the steel industry worldwide is significantly changing following a social and political trend and the requirement for fossil-free energy generation and industrial production to be in accordance with the world climate targets committed to in relation to the decades leading up to 2050. Iron and steel production is one of the major polluters of climate changing greenhouse gases; it must change to renewable primary energy sources and the use of climate-neutral reduction agents. Because it is very obvious that carbon, as the main component for steel strength properties, cannot be eliminated totally from the steel production process, the question arises of where a "zero carbon" approach can lead? This paper will review the ongoing success story of the LD-process, discuss the recent technology advancements, and give an outlook on the future role of the process in the steel industry.

**Keywords:** oxygen steelmaking; LD-process; BOF; zero carbon; process technology; history; advancements

## 1. How It Started

At the end of WWII, Austria was occupied by the allied forces: the USA, UK, France, and the Soviet Union. All industrial assets had been heavily damaged by air raids during the war. In 1946, the assets were returned to the newly formed Republic of Austria, and reconstruction under the framework of the Marschall Plan started at VÖEST-Linz and ÖAMG-Donawitz. In June 1947, the first blast furnace in Linz was blown-in, and steel production restarted with a repaired open-hearth furnace. In 1948 the new government launched the "Iron and Steel Plan", which allocated the production of flat steel to Linz and the production of long products to Donawitz. The plan included starting the production of finished products in a heavy-plate mill, restored after air raids and demolition, and building a new slabbing and hot-strip mill in Linz.

In the meantime, all important players in the LD-history: T. Suess, H. Trenkler, H. Hauttmann, and H. Weitzer had arrived back in Austria from their former employments in the German steel industry [1]. From the very beginning, it was obvious that basing the future flat production in Linz on open-hearth technology was too expensive to succeed, but

time was running out because the new facilities were already under construction. Based on this conclusion and knowledge about trials carried out during the war to produce steel from hot metal by oxygen refining, the team searched for other activities in this field in Europe [2].

When they heard about successful trials with oxygen purification at the von Roll steel mill in Gerlafingen, Switzerland, directed from Prof. R. Durrer, the Austrian engineers visited the plant for a technical exchange. After this visit, the decision was taken to develop the process [3].

In May 1949, a conference on oxygen metallurgy was held at the Montanuniversität, Leoben, with participants from Austria, Germany, and Switzerland. A division of tasks in developing oxygen for steel production was agreed upon between von Roll's Gerlafingen, Mannesmann AG, VÖEST, and ÖAMG. In Gerlafingen, oxygen refining would take place in the EAF, in Huckingen, bottom injection would be carried out in the Thomas converter, in Donwitz, oxygen was used for ore reduction in a shaft furnace and for obtaining a manganese-rich slag for alloying purposes, and in Linz, the refining of hot metal with pure oxygen was to be developed [4].

Trials began in Linz on 2 June 1949. In a two-ton converter with vertical blowing of oxygen onto a hot metal melt, the breakthrough was achieved on 25 June after initial failures. Hot metal could be refined in a reproducible manner into steel with lower oxygen pressure and a greater distance of the blowing lance from the melt. The steel could be rolled without any problems, and the testing laboratory managed by H. Hauttmann judged the quality of the new steel to be excellent.

In Linz, the trials were systematically continued. In parallel, a refining station for blowing oxygen in a 15-ton vessel was built on the outer wall of the steel plant, which was commissioned with the first batch on 2 October 1949; see Figure 1 [5]. These trials subsequently clarified questions of the large-scale feasibility of the process, its economy, and the new steel's service properties. It was possible to roll plates of excellent quality from 11-ton slab blocks [6].

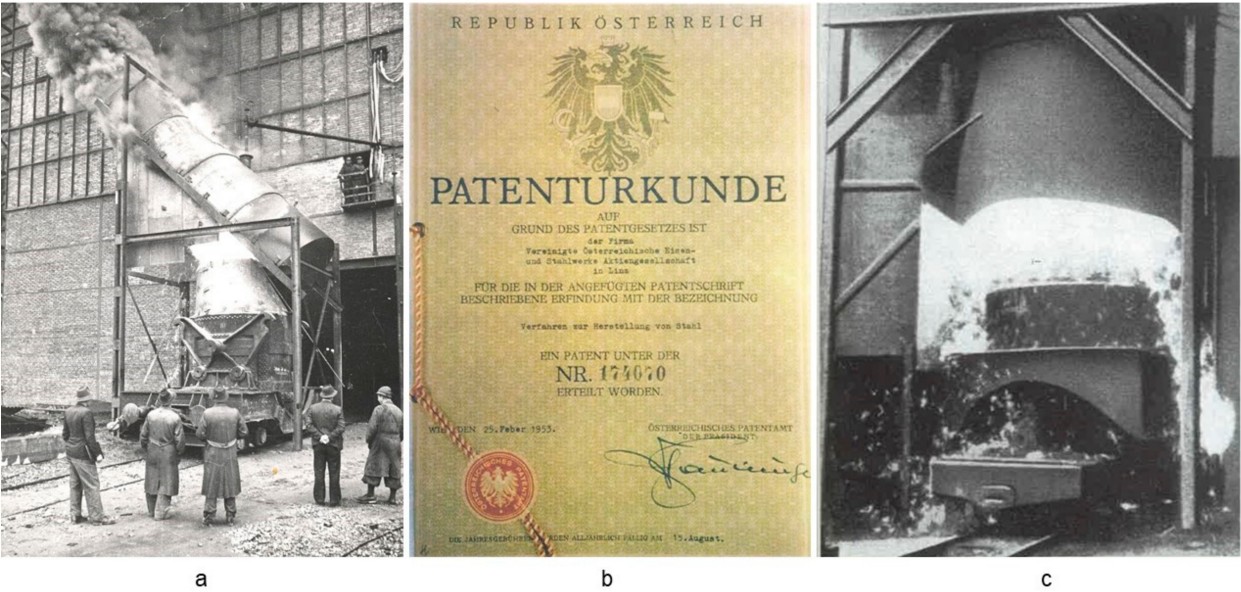

**Figure 1.** First blowing trials with 15-ton vessel in Linz (**a**), LD-Patent from February 1953 (**b**), and blowing trials with 10-ton ladle in Donawitz (**c**), Reprinted with permission from Refs. [5,7,8], 2022, Voestalpine AG.

Very quickly after the first trials on 9 December 1949, VÖEST decided to build an oxygen-blowing steel plant. In October 1950, the design plans for the new steel plant were finalized. The equipment was purchased from GHH, Germany. The new steel plant with two 30-ton converters was built adjacent to the existing open-hearth steel plant. The steel

plant layout shows a pear-shaped converter and the charging and tapping equipment in the same hall aisle, which are the characteristic features of a Thomas steel plant. The investment was financed from the Marshall Plan funds established in 1947.

In Donawitz, oxygen blowing tests were also carried out in 1949. Whilst blowing on hot metal in a 5-ton ladle to refine high manganese hot metal, it was recognized that de-carburization (De-C) had taken place simultaneously and steel had been produced. The experiments were systematically extended for steel production and testing. By exchanging information with the Linz plant, it was possible to confirm with each other that vertical, relatively gentle blowing with a greater nozzle distance from the melt was the way to success. This was different to the opinion of Hellbrügge and Durrer in Gerlafingen, who still maintained that hard and deep blowing of the oxygen is necessary to reach sufficient De-C [3].

The tests at Donawitz were continued in a larger blowing stand, a 10-ton ladle with a conical hat on top and lined with refractory, and the oxygen supply given from a battery of cylinders connected in series. After about 30 trial melts and thorough testing of the steels produced, it was also clear to Donawitz that the expansion of steelmaking should be carried out using the blowing process with pure oxygen. In June 1950, Donawitz engineers designed a so-called SK steel plant (German abbreviation for: "Sauerstoff-Konverter"—"Oxygen converter") for an annual production of 450,000 tons. The steel was first called PUROX, and from 1953 onwards it was branded SK steel, similar to its production process. The new layout of the SK-steelmaking plant became the prototype of all future oxygen steel plants [7].

The rotationally symmetrically converter is at an elevated level in the central nave of the steel plant building. There are also located bins for additives and the blowing lance equipment. The hall for hot metal and scrap charging is on one side and the tapping and casting hall is on the other (see Figure 2). The steelmaking equipment for Donawitz was supplied by DEMAG, the oxygen generation plant by LINDE. In 1950, both VÖEST and ÖAMG applied for the first patents on their LD and SK processes, respectively; Figure 1 [8].

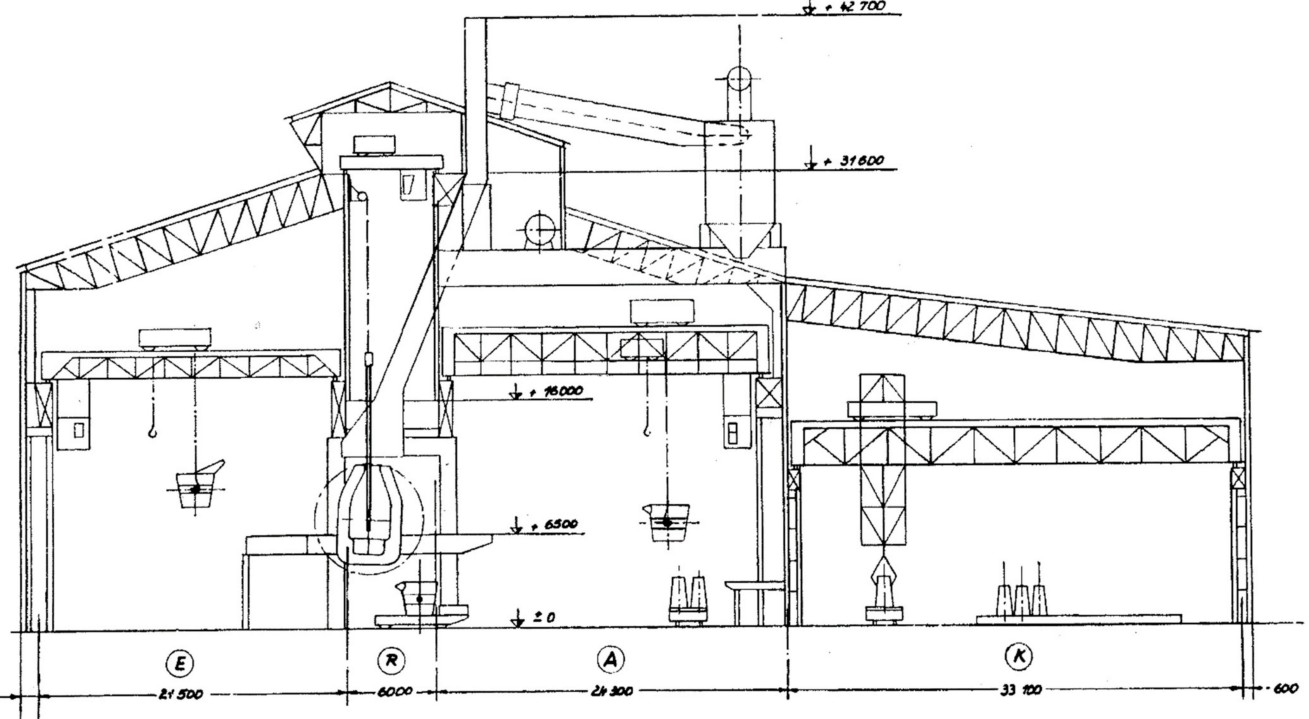

**Figure 2.** Section view of the SK steelmaking plant of ÖAMG in Donawitz in 1953. Reprinted with permission from Ref. [8], 2022, Voestalpine AG.

While the two steel plants were under construction, blowing tests and steel quality investigations were systematically continued and gained international interest around the globe. News of the oxygen blowing tests soon spread. Already in July 1951, engineers from Nippon Kokan K.K. visited the Donawitz and Linz plants. The Japanese were then among the first to take out a license for the patented new process and modernize their steel production accordingly [9].

After two years of construction, the world's first LD steel plant was ready for commissioning in Linz. On 27 November 1952, the first heat was blown in converter No. 2. On 5 January 1953, the new LD steel plant and the new slabbing and wide hot-strip mill were officially opened by Austrian President T. Körner [1,5].

In Donawitz, the commissioning of the oxygen steel plant with two 30-ton converters was delayed until 22 May 1953 due to the late delivery of materials for steel construction. At that time, almost 100,000 tons of LD steel had already been produced in Linz. The process that was developed in parallel in Linz and Donawitz exceeded all expectations in terms of performance, economy, and the quality of the steel produced.

Despite the proven outstanding properties of LD steels, there were major obstacles to its introduction into international standards and codes. As a "converter steel", it was initially placed in the group of blast refining steels and rose above the Siemens-Martin steel grade. Therefore, in Austria, the converter was called a crucible because crucible steel denoted the highest quality level.

In March 1952, Brassert Oxygen Technik (BOT) was founded as a Swiss stock corporation with Zürich headquarters. In the same year, VÖEST and ÖAMG signed a BOT shareholding contract. The two steel enterprises contributed their patents and know-how for the shares. From 1953 onwards, BOT concluded license agreements with almost all steel-producing countries globally. The licensees could use all the patents held by BOT and the help and know-how of VÖEST or ÖAMG in commissioning the plants. However, they had to contribute their own developments to the BOT, which were then available to all licensees. This clause served both the old and the new licensees and brought about a dynamic further development of the new steelmaking process.

Figure 3 shows the world steel production and the process share since the industrial revolution in 1860. The LD-process (shown as Basic Oxygen in Figure 3) became the globally dominant crude steel production technology, starting in 1960. After the initial metallurgical success, another major achievement of the technology was to remove dust from the process waste gas, the notorious red smoke. In 1958/1959, this was successful: in Linz, with the wet dedusting process, and in Donawitz, with the electrostatic dry dedusting process concept. This was another significant step toward the worldwide adoption of the LD-process [6].

With the developments of ARBED-Luxembourg and CRM Belgium (LD-AC-process) and the French research institute IRSID (OLP-process), the processing of phosphorus-rich hot metal also became possible in the late 1950s [9]. By 1962 there were already 80-ton LD-AC converters producing steel with much higher output and quality than the traditional Thomas process. The invention of the taphole at the transition to the conical converter hat also dates from this period [9]. This enabled clean tapping and safe deoxidizing and alloying of the melt in the steel ladle. The sequence time could thus be significantly reduced. With the increasing tapping weight of LD converters (100 tons in 1960, 200 tons in 1962, 300 tons in 1964, 400 tons in 1968) and the associated process and equipment improvements, the performance of LD-steel plants increased enormously [12]. The ongoing improvement of the refractory lining of the converters also contributed significantly to this. Lining with magnesia bricks was common practice in Donawitz from the beginning. The Linz converter had a dolomitic lining. Combination lining with doloma, doloma-magnesia, and magnesia, optimizing the economic and productional aspects, has been a main trend over the decades [13].

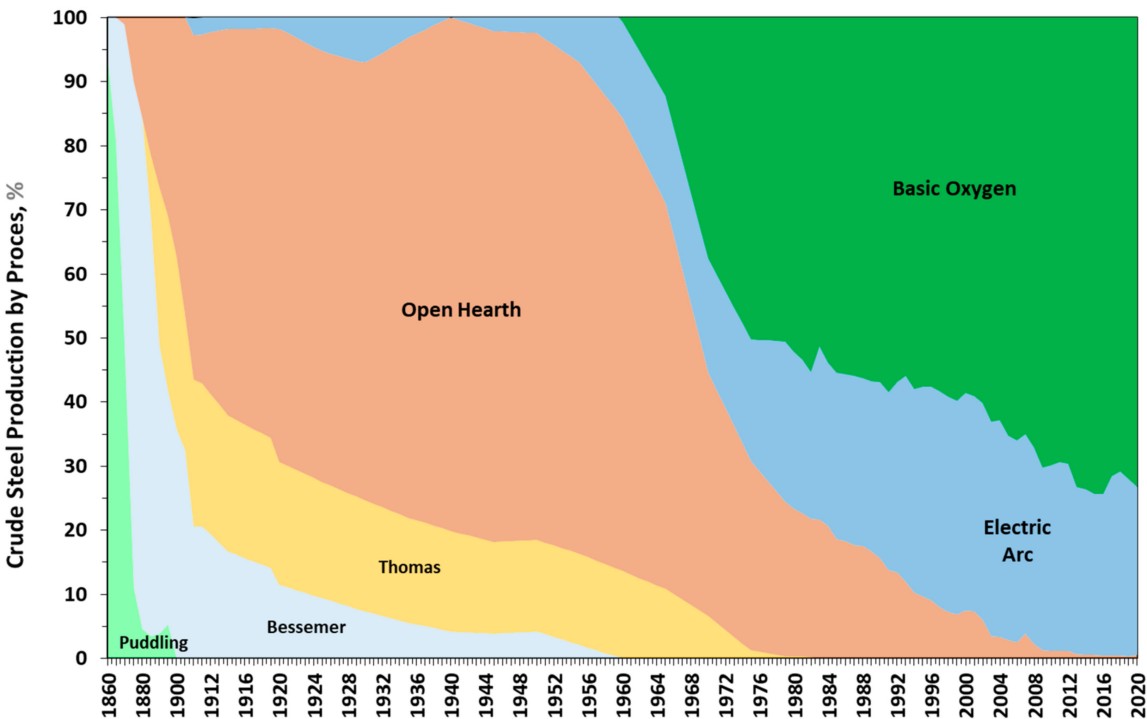

**Figure 3.** Development of the world steel production and the share of steelmaking technologies since 1860 [10–12].

With the increasing converter heat size, from approximately 100 tons tapped weight, the original single-hole Laval oxygen nozzle was replaced by multi-hole nozzles to ensure an appropriate oxygen supply and, at the same time, soft blowing for slag formation.

Measurements in the off-gas duct for the flame temperature were used for process control. With the development of the OBM-process (OBM stands for Oxygen Blowing Maxhütte) by Maxhütte in Sulzbach-Rosenberg, who started up the first converter with oxygen injection through the bottom in 1968, further essential developments were also initiated for the LD-process. As early as the beginning of the 1970s, bottom purging elements were developed with the refractory suppliers of the converter lining, allowing inert gases (Ar, N$_2$) to stir the melt [14]. Combined oxygen blowing via the lance from above and from the bottom nozzles below was also developed [15].

Despite the decline in steel demand as a result of the first oil shock in 1974, 245 converters were newly commissioned in the 1970s, and decisive innovations were introduced in LD steel plants. Secondary dedusting and combustion-free flue gas routing improved the environmental friendliness of the LD-process, and the now possible co-gas-recovery improved the energy balance [16]. Continuous off-gas analysis; sublance technology for measuring temperature, carbon content, and oxygen activity; and continuous weight measurement of the converter, etc., in conjunction with physicochemical process models, allowed automation of the process and more precise adjustment of temperature and crude steel analysis. The development of slag retention systems and electromagnetic early slag detection also made a significant contribution [17,18].

Leading at this time were the Japanese steel plants, where hot metal pretreatment (de-siliconization and de-phosphorization, in addition to the generally used de-sulfurization) and hot metal refining with low slag volume were developed. In summary, by the end of the 1970s, the LD-process as we know it today had been developed, including the interaction of hot metal pretreatment, ladle metallurgy, and continuous casting to produce the highest quality steel. The application of the various process variants developed is shown in Figure 4. The dominant variants are straight-top blowing (LD) and combined top-blowing and bottom stirring with inert gas (LD-BS). The bottom-oxygen blowing process

variants (LD-OB, KOBM, and OBM) became fewer over the decades. The main reason for this is that it is more sophisticated to run the process with bottom-oxygen tuyeres than to run a simple top-blowing process.

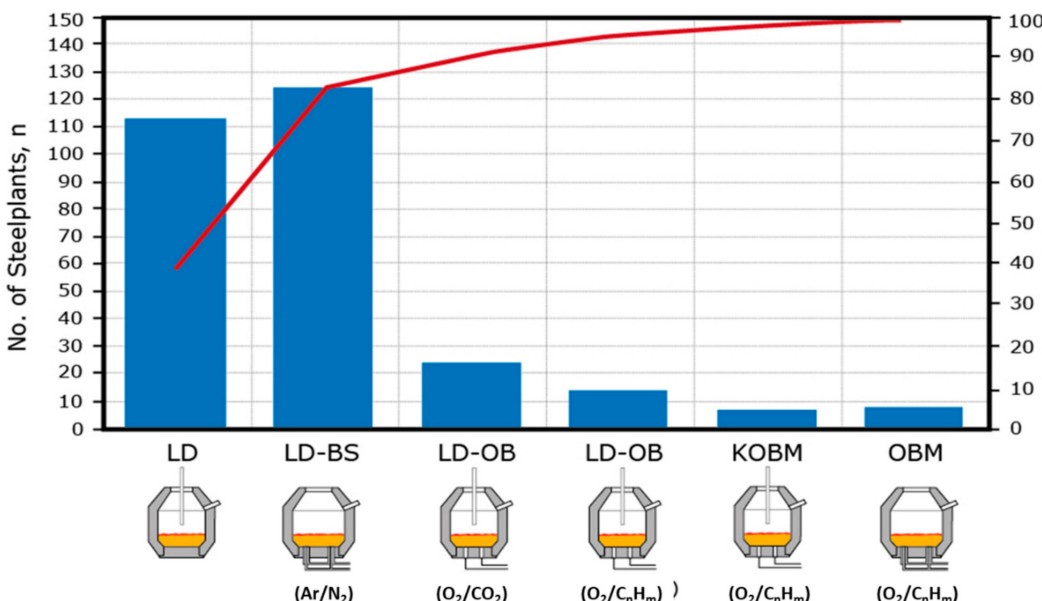

**Figure 4.** LD-process variants applied worldwide (2018) (the bars represent the number of installed converters and the red line represents the cumulative share of the six converter types).

## 2. State-of-the-Art in Technology

The past decade has seen numerous innovations in both plant technology and process control across the entire LD-process. A graphical representation of the technological innovations implemented over the past decades is given in Figure 5. The focus was on improved plant availability and process stability, mainly driven by the increasing quality requirements for the blowing process itself. Furthermore, the requirements for the post-treatment to produce high-quality steel grades increased [19,20].

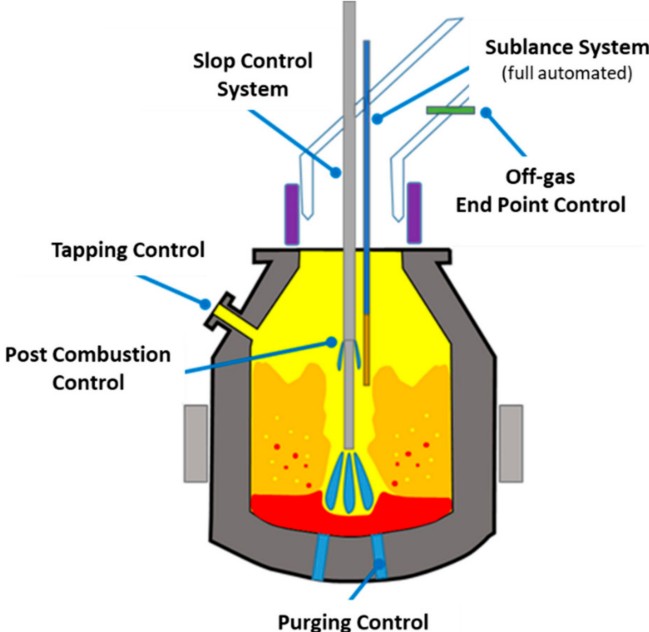

**Figure 5.** Overview of the technological innovations in the LD converter.

### 2.1. Converter Equipment

In the case of new construction or enlargement of the converter vessels themselves, models for the optimal design of the converter geometry support many plant engineers. The increase in the service life of the converter vessel shell is improved, for example, by additional cooling systems in the hat area or the trunnion ring itself (ring gap systems). High-strength steel materials with correspondingly high heat resistance are increasingly used, even if their processing and repairs are associated with significantly higher costs (temperature control) to avoid cracking. Innovative trunnion ring systems whose task is to create a secure connection between the ring and the converter vessel itself are being used and contribute to increasing the service life and operational safety. [21] Removable converter bottoms enable rapid dismantling and assembly of the vessel bottom and shell. In combination with modern brick lining machines, the relining times of a converter could be reduced [22]. The use of fast frequency-controlled tilting drives with corresponding torque for moving the high masses of vessel, refractory material, and contents goes hand in hand, whereby increasing automation requires an ever-higher accuracy of the rotation angles.

### 2.2. Blowing Process

The blowing lance geometry was optimized for the larger converter vessels based on sophisticated CFD models in combination with water model tests [23]. Post-combustion distributors (PCD), through which a portion of the oxygen is injected via fine ring channels along the lance, make it possible to reduce lance skulling and increase the operating time of the blast lance by post-combustion of the converter gas, Figure 6 [24–26].

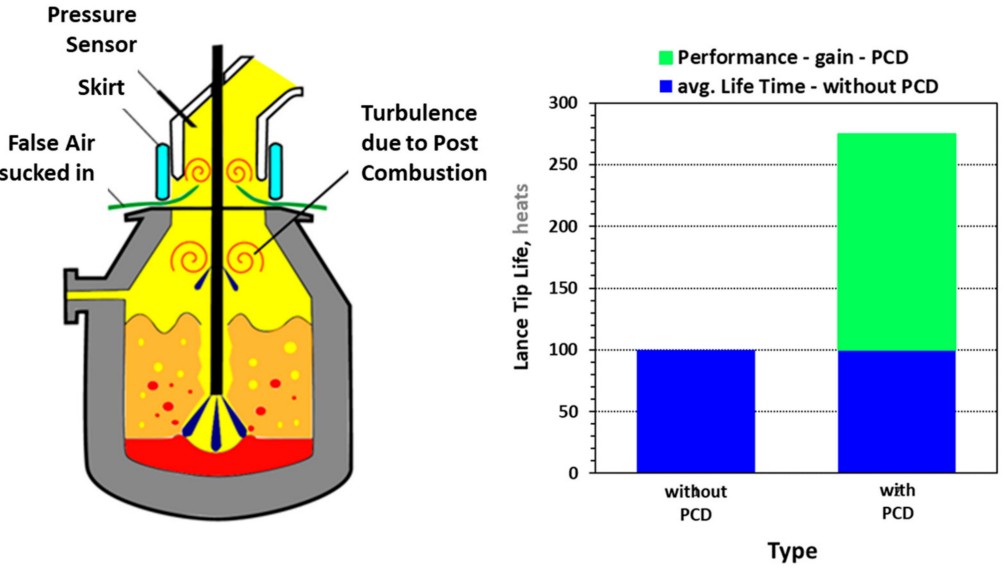

**Figure 6.** Advantage of PCD-units for increased lifetime of oxygen-blowing lance and lance tips [26].

At the same time, this technique reduces the skulls in the top cone area of the converter. A separate controlled system for the post-combustion unit (dual flow post-combustion (DFPC) lance) enables additional heat to be recovered in a targeted manner. It thus allows the scrap rate to be increased, but in return reduces the amount of energy to be recovered during electricity generation. Increasing the scrap rate at the converter is currently one feasible possibility to reduce the $CO_2$ emissions of the integrated route. In addition to the DFPC lance, lances for scrap preheating also show potential here [27].

Depending on the availability of hot metal and its chemically bound heat (carbon, silicon, manganese, and others), the hot metal ratio in the ferrous charge differs in variable proportions between 72 and 95% of hot metal. An overview of the hot metal ratio for different BOF steelmaking plants in the Americas, Europe, and Asia is presented in Figure 7. The dots marked with CSA and VASL are the extreme cases for medium-sized converter

steel plants. In case of intensive hot metal use, a considerable number of additional coolants (iron ore, sinter, iron dust, other) is necessary, which can lead to an unsteady blowing behavior. The same effect occurs with very low hot metal ratios, because here the addition of heating agents (ferrosilicon, coke, other) interrupts the de-carburization during the refining. A balanced heat balance or correspondingly high reaction volume in the converter should be aimed for, if possible, to avoid slopping. For the early detection and reduction of slopping, numerous measuring systems are in operation or in test use, which predict the blowing process's stability, based either on acoustic signals or vibration measurements [28].

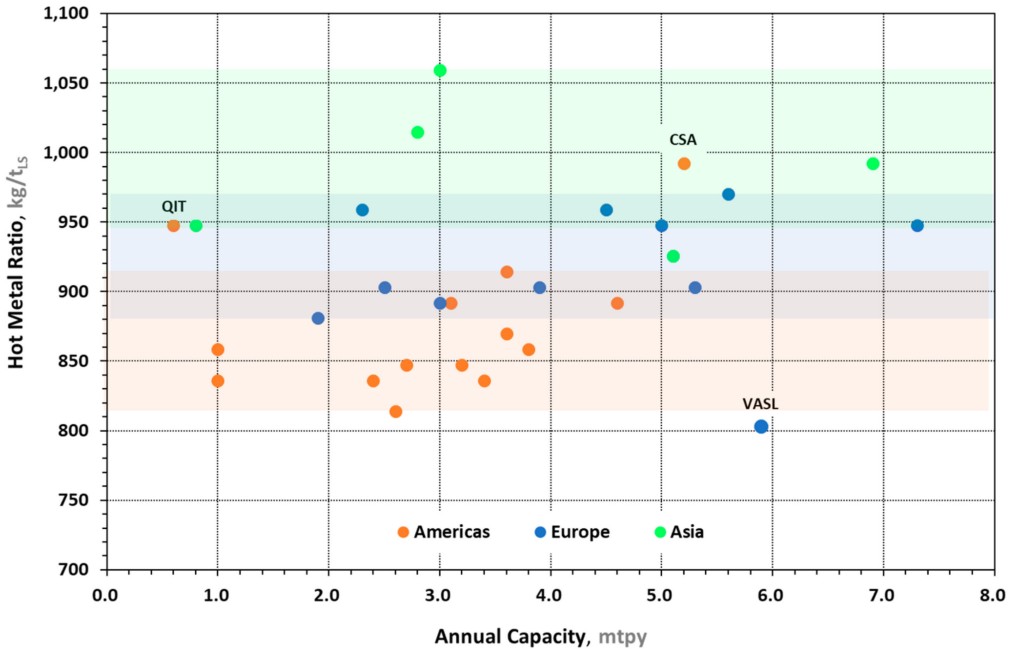

**Figure 7.** Overview of hot metal ratio of steel producers worldwide.

## 2.3. Bottom-Purging

The arrangement and availability of the bottom purge, which is indispensable for achieving the lowest phosphorus content, were optimized. High purging performance with high availability up to the end of the lining campaign of up to more than 4000 melts are demanded from the purging systems. An essential specification is that wear of the purging blocks and their surrounding refractory material should coincide [23]. The use of multi-hole plug (MHP) purging blocks compared to single-hole plug (SHP) blocks is becoming increasingly common. The corresponding upgrading of the purging gas control station to supply the purging gas media argon and nitrogen in sufficient quantities is mandatory. Purging with constantly high purging rates ensures the necessary good mixing of the melt and promotes the steel/slag reaction. Separate control of the individual plugs to generate additional effective turbulence and homogenization of the steel has been suggested in some cases [29].

## 2.4. Endpoint Detection

For the predetermination of the carbon content at the end of the main blowing process as well as the final blowing temperature and the determination of the final blowing point, powerful process models are used. They are combined with waste gas analysis systems using spectrometers or laser systems and sublance systems (Figure 4). The main advantages are an increase in efficiency in the quality fulfilment, the steel yield, and the protection of the refractory material with an overall significant cost reduction [24,30,31].

Sublance systems were innovatively adapted with robot technology to ensure availability on the one hand and to completely take over the manipulation of the large probe bodies from the employee on the other (see Figure 8). Camera systems are also combined

with sophisticated system logic to control the entire manipulation and measuring process. This provides additional safety for employees by eliminating the risky steel sampling from the converter [32,33]. The endeavor regarding an on-site steel analysis without the necessary preparation of the sample to analyze the steel composition is already partly in use. For this purpose, container laboratories with fully integrated laboratory technology, including automatic manipulation with high sample throughput and analytical accuracy, are installed [34].

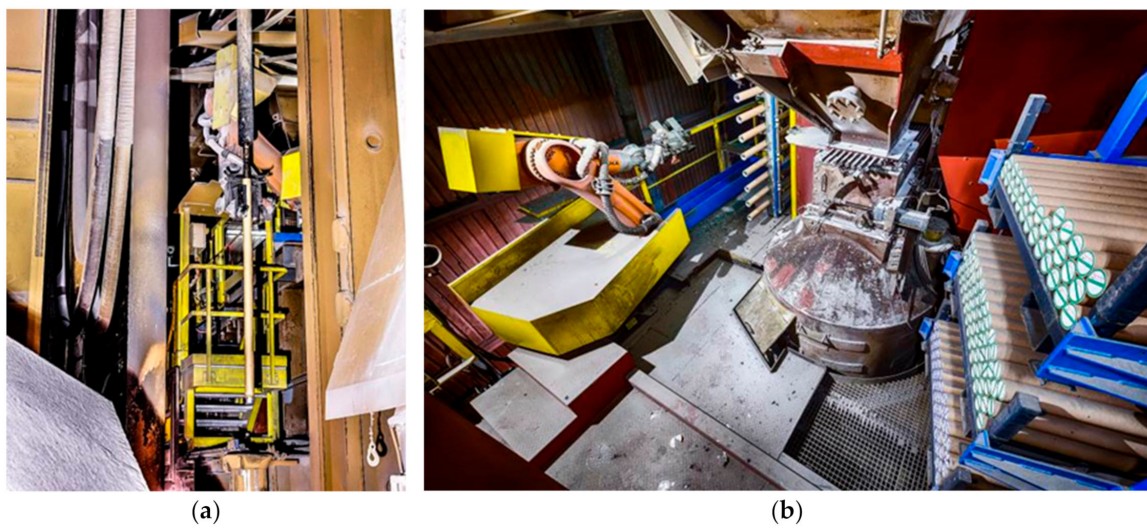

(**a**)                              (**b**)

**Figure 8.** Installation of the new sublance probe manipulation system. The robot attaches the probe to the sublance (**a**); robot with the magazine for probe storage (**b**) [32].

### 2.5. Tapping

The primary objective during crude steel tapping from the converter is the prevention of the carry-over of the phosphorus-rich LD slag to the steel ladle. While ball and dart systems based on gravity are still used in some LD steel plants, most plants have switched to pneumatic closure devices [35]. Infrared-based camera systems have replaced electromagnetic detectors (EMLI) due to their low availability (see Figure 9). By detecting the different emission spectra of steel and slag, an almost slag-free tapping (2–3 kg slag/ton crude steel) can be achieved with pneumatic closure units (slag stopper hammer) despite harsh conditions caused by smoke and dust. New camera systems allow higher resolution, higher availability, and significantly smaller size. This leads to a considerable reduction in maintenance costs.

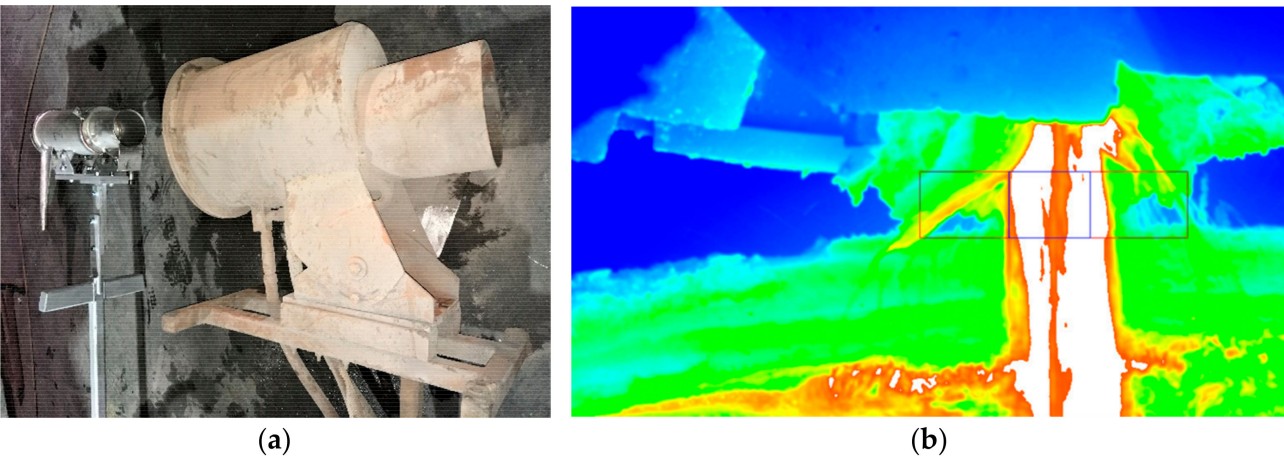

(**a**)                              (**b**)

**Figure 9.** (**a**) Slag stopping systems with (**b**) infrared slag detection latest generation.

In recent years, the development of semi-automatic or automatic tapping technology has been an endeavor for steel manufacturers. The aim is to achieve a high level of system availability, with the tapping process merely being monitored by the operating personnel. The basic conditions for this are a suitable, exact control of the tilting drives as well as the necessary video equipment, including image evaluation. The positioning of the LD converter under the condition of the maximum steel bath level height above the tapping spout and the avoidance of the converter slag overflowing the converter mouth are mandatory requirements. New generation, model-supported camera systems offer a high degree of reliability in this respect, even in harsh environmental conditions [36,37].

### 2.6. Off-Gas Cleaning

In many LD steel plants, off-gas purification and utilization are carried out by waste heat boilers in which steam is generated by the hot converter gas (1600 °C), which is primarily utilized in the steel plant (vacuum systems, blast furnaces, hot water preparation). In the evaporative cooler, water and steam are used as atomizers to remove the coarse dust from the exhaust gas, which is then further cooled (200 °C). Fine dust is separated from the dry gas in electro dry filter systems. Sensors monitoring for optimum dust separation and avoidance of skull formation in the exhaust system have been used for years and are also used for process control. The dry, coarse, and fine dust is further processed by hot briquetting, pelletizing, or granulation. Inline zinc measurements are used to control the extraction of zinc-containing byproducts. The granules (>10% Zn, ~15 moisture content) offer higher durability, especially when transported to recycling plants. The low Zn fractions (<10%) of dry dust briquettes are used in the LD-process as additional cooling agents during the main blowing process.

### 3. Worldwide Operation Practices Applied

### 3.1. Straight Blow Practice

Over time, the oxygen steelmaking process was always adapted to the local conditions and availability of raw materials. In Europe, in the late 1960s, the second generation of LD steel plants was built, and the process was modified during the next decade to the first LD top-blowing converter and later LD with bottom stirring with inert gases (Ar/$N_2$). At the same time, the raw material supply for the new blast furnaces was switched from domestic low-Fe sources with high phosphorous content to Fe-rich sources with low phosphorous content. The low phosphorous contents in hot metal (0.050–0.090%) in combination with the low silicon contents (0.30–0.60%) allowed the operators to focus on a "straight blow" operation practice, which is characterized by simultaneous metallurgical reactions during a single blow. As illustrated in Figure 10, the sequence is opened by charging scrap and hot metal and finished by steel and slag tapping. The time between two heats is used for refractory inspection and maintenance. The preventive maintenance applied is basically slag coating and gunning. Productivity and metallurgical results are more of a focus than the lining campaign life. This operation practice is mainly applied in the European LD steel plants.

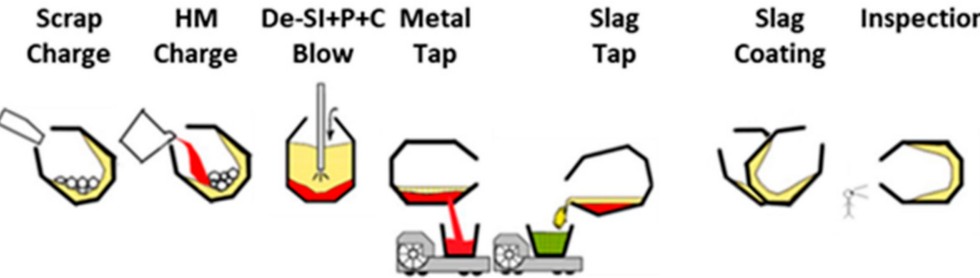

**Figure 10.** Straight-blow practice (blow-stir practice).

### 3.2. Blow-Splash Practice

In North America, at the beginning of the 1980s, slag splashing technology was developed to increase the vessel lining life significantly and to drastically reduce the downtime for relining [38–42]. The technology is characterized by blowing the remaining slag after tapping to the furnace walls (refer to Figure 11). The blowing lance is operated with nitrogen and is positioned at a low height distance to the furnace bottom. The resulting operation standard is known as "blow-splash" operation practice. It requires a special slag composition with high (MgO%) enrichment for supersaturation, low (FeO%)- and (Al$_2$O$_{3\%}$)-contents (preferably <15% for (FeO%) and <3% for (Al$_2$O$_{3\%}$)), and low tapping temperatures (<1.660 °C) to guarantee low viscosity and sufficient sticking properties [42,43]. Specially designed splashing lance tips and independent lance cartridges are used to blow the slag to the walls in different heights depending on the area to be protected [44]. The most critical areas to be covered with protecting slag layers are the trunnions, which are difficult to reach with other maintenance methods applied [40].

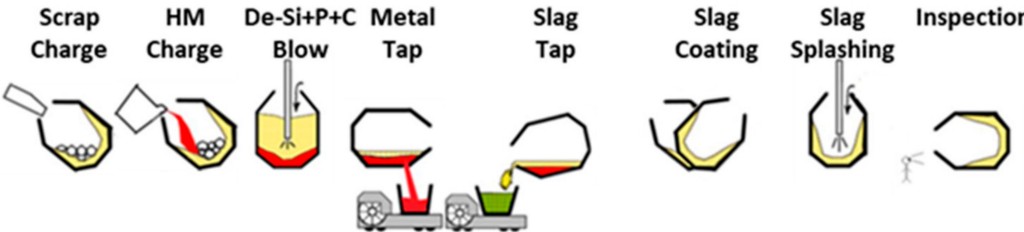

**Figure 11.** Straight-blow with slag splashing practice (blow-splash).

The achievements, on the one hand, were noticeable with campaign lives of >20,000 heats, resulting in downtime savings of 9–13 days for relining per lining campaign [45,46]. On the other hand, the additional costs for MgO-flux are inevitable, and they include processing of the increased slag volumes, poorly processed slag material properties, and additional refractory gunning material, which is used in the 2nd half of the campaigns to prevent failure when the residual brick thickness cannot be measured anymore because of the protection layer sticking on the bricks. Other problems caused by excessive slag splashing are:

1.  Blowing lance skulling;
2.  Skulling of the vessel mouth;
3.  Blocking of bottom stirring plugs/tuyeres (by bottom built-up);
4.  Limited efficiency of the hot face residual brick thickness measurement by lasers;
5.  Higher sensitivity for slopping/spitting;
6.  Cycle time losses for splashing, inspection and refractory maintenance;
7.  Yield losses (change in vessel profile);
8.  Poor metallurgical slag properties for de-phosphorization.

It must be concluded that excessive slag splashing is beneficial for plants with the availability of low-phosphorous hot metal ($p$ < 0.050%) and low tapping temperatures (<1.660 °C), moderate-low productivity, and capacity/allowance for slag disposal. In most regions of the world, these conditions cannot be guaranteed. Slag splashing should be applied to complete the refractory maintenance measures but to a limited extent. The focus must be given to the metallurgical results of the process.

### 3.3. Double-Slag Practice

The "double-slag" operation practice was developed to achieve ULP (Ultra-Low Phos) conditions for either unstable hot metal conditions or high phosphorous contents in hot metal. As shown in Figure 12, in this practice, the high de-phosphorization capacity of low temperature and low basicity slags during or after the de-siliconization phase is used to remove the slag in a slag-off break after 5–7 min blowing at still high carbon contents. In the 2nd phase of the blow and by charging of fresh lime flux, a high-speed de-carbonization blow without danger of slopping is applied to finish the blow.

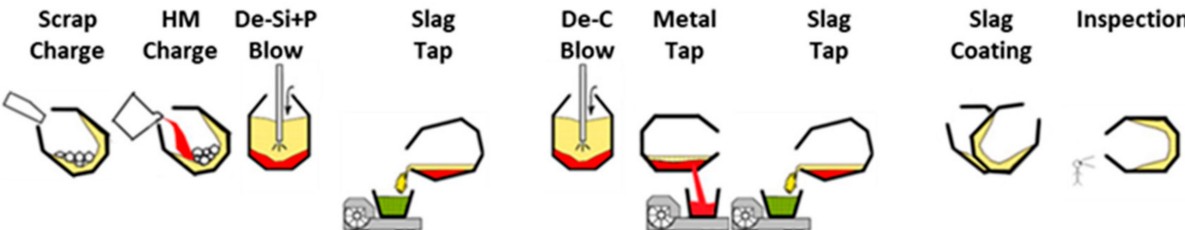

**Figure 12.** Double slag practice in a single vessel (double-blow-stir).

The "double-slag" practice is not new; it is a modified version of the LD-AC/ORP process [47–49], which was applied in the early years of the LD-process operation. It was developed in Europe where domestic iron ores with high phosphorous contents were processed in Germany, France, Belgium, and the UK. For this purpose, the steel plants were equipped with lime injection through the blowing lance, but these types of equipment were no longer installed in the second generation LD steel plants in the late 1960s because the hot metal quality changed, as explained earlier. In 1985, the German steel producer association investigated the double slag practice in a paper but concluded that it was not required for the European hot metal conditions [50]. The plants then focused on the optimization of the straight blow practice.

In Asia (Japan, South Korea, and Taiwan), the hot metal conditions were always different and were characterized by high hot metal phosphorous contents ($p > 0.120\%$). This condition results from the iron ore mix used to reduce hot metal costs. High phosphorous iron ores are available worldwide in higher resources than premium qualities mined in North America, Brazil, and Australia. Additionally, Japanese steel producers in particular were always focused on the quality advantages of ultra-low phosphorous steel grades ($p < 0.010\%$) in combination with high alloy contents, which are difficult to maintain on a stable average in a straight blow operation. Because these grades are usually produced based on strictly defined orders, the potential of statistical process control and grade shift is not applicable.

For this reason, in the early 1980s, hot metal pretreatment either at the blast furnace runner or torpedo cars by injection of flux and scale was studied and developed [51]. At the end of the 1990s, the double slag process using two different vessels, the "2-vessel-double slag" operation practice, was introduced (refer to Figure 13). In this practice, metal and slag are both tapped-out after the first blow for de-siliconization and de-phosphorization. The metal is charged into the high-speed de-carburization vessel for a quick blow with fresh lime flux.

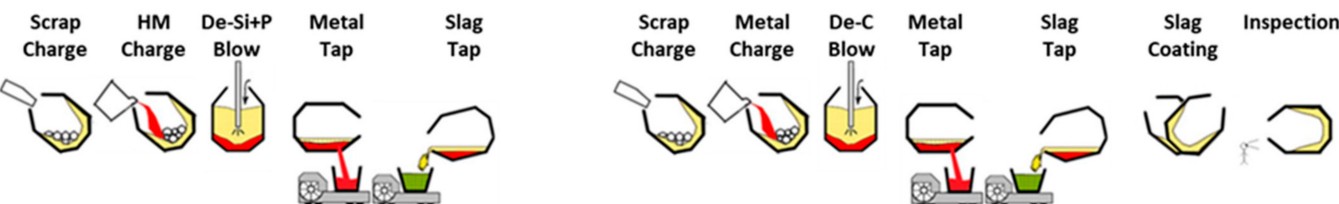

**Figure 13.** Double slag practice in two separate vessels (2-vessel-double slag).

By applying this technology, both process steps could be optimized individually for improved metallurgical and productivity results [47]. In some plants, the tap weight is varied between the first and second blow, and the scrap charge is split to increase scrap rates and reduce cycle time.

Another topic that can be handled sufficiently in the two-vessel operation is the separation of the slag from the two process steps. The slag from the first blow, rich in silica and phosphate (low basicity), is tapped, cooled down, and processed for secondary usage (road and dam construction). Because of its low basicity, this slag does not need a

"weathering" treatment, as is commonly applied for LD-slags to eliminate free CaO and MgO from the slags for stability purposes. The high basicity, high (FeO%) slag from the second blow is then also cooled down and processed for cold charging into the first blow or even hot recycled into the first-blow vessel. Another option is to use the second-blow vessel of the first heat as the first-blow vessel of the second heat. The vessel lining life in the split process route is high, up to 10,000 heats on a lining, although slag splashing and (MgO%)-enrichment is not applied. The reason for this is the low-temperature operation in the first blow, with the initial lime-undersaturated slags and lime-oversaturated operation at the higher temperatures during the second blow. Operation at lime saturation reduces the MgO solubility of the slag and thus reduces the refractory wear.

Today, depending on the overall plant design and product mix, all major steel companies in Asia have installed new combinations of de-sulfurization stations (KR-process) with de-siliconization and de-phosphorization LD converters in front of their existing steel plants in separate aisles or buildings [52–56]. Depending on the steel grade requirements, a mix of "straight-blow", "double slag"or "2-vessel-double slag" operation is applied. A typical differentiation for an initial hot metal phosphorous content of 0,120% could be:

(a) Straight-blow: for final [%P] 0.015–0.025% (47%): 35 min/heat; improvement of slag volume by using recycled LD-slag/ladle slag in charge, in case of liquid recycling temperature, and FeO content must be decreased.

(b) Double-slag: for final [%P] 0.015–0.025% (51%) and 0.008–0.012% (2%): 45 min/heat operation time.

(c) 2-vessel-double-slag: for final [%P] is 0.005–0.008: 1st heat: 34 min/heat + 2nd heat: 28 min/heat = 62 min/heat; increased Fe-yield.

With the consequent application of hot slag recycling in Japan and South Korea, the total lime flux consumption was reduced to <30 kg/tons and slag recycling/secondary usage reached up to 100%. Besides the deposit/landfilling permission issue, a lack of flux sources in these countries is the driver for extreme activities. In many countries, slag is still considered as emission causing, hazardous waste.

## 4. LD-Process towards Reliable, Safe, and Sustainable Steel Production

The Paris Agreement adopted in 2015 defines global targets for reducing greenhouse gas emissions. The global steel industry expressed its commitment to reducing the $CO_2$ footprint from its operations and the use of its products, including by-products. The World Steel Association defined three main elements for the implementation:

(a) Reducing the impact of steel production;
(b) Efficiency and the circular economy;
(c) Developing advanced steel products to enable societal transformations [57].

The LD-process is a highly optimized and mature technology. However, research and development have been carried out over the last few decades and are ongoing to enhance the reliability, safety, and sustainability of steel production in the integrated route. The main goals for enhanced sustainability are reducing $CO_2$ emissions and the increased usage of the by-product slag and dust as secondary raw materials.

### 4.1. Increase of Scrap Rate

A strong means of lowering the $CO_2$ emissions of the integrated steel production route is the increase of the ratio of scrap to hot metal in the charge. One measure to achieve this is to increase the post-combustion rate. The application of post-combustion, i.e., oxidation of CO leaving the metal bath to $CO_2$ in the gas phase above the melts, to increase the scrap rate is not new. The historical development of this technology began with the introduction of the OBM process at Maxhütte in Unterwellenborn, Germany, in 1968. By installing additional oxygen nozzles in the converter hat (2–4 nozzles depending on the converter size), Brotzmann [58] et al. succeeded in increasing the scrap rate in the OBM process by 5–10%.

Gruner et al. published the first operation results for the scrap rate increase combined with the design of converter vessels and blowing lances in 1984 [59,60]. In the following years, the effect of post-combustion on the scrap rate was investigated by different research groups. Hirai [61] summarized his results:

(a) The scrap rate can be increased by 0.34% with a 1.0% increase in the degree of post-combustion.

(b) With a degree of post-combustion between 10 and 20%, around 70% of the heat of the reaction can be transferred to the metal bath and the slag.

In the following years, attempts and trials to increase scrap rate with post-combustion were reported from LD steel plants in North and South America [62–64]. The most recent publications are from Russo et al. (2004) [65], Balkos et al. (2005) [66], and Cameron et al. (2008) [67]. They examined the Cold Shroud © technology; the principle of this technology is shown in Figure 14. Oxygen from several secondary nozzles, arranged around the main nozzles in the lancehead, causes the "shrouding" effect.

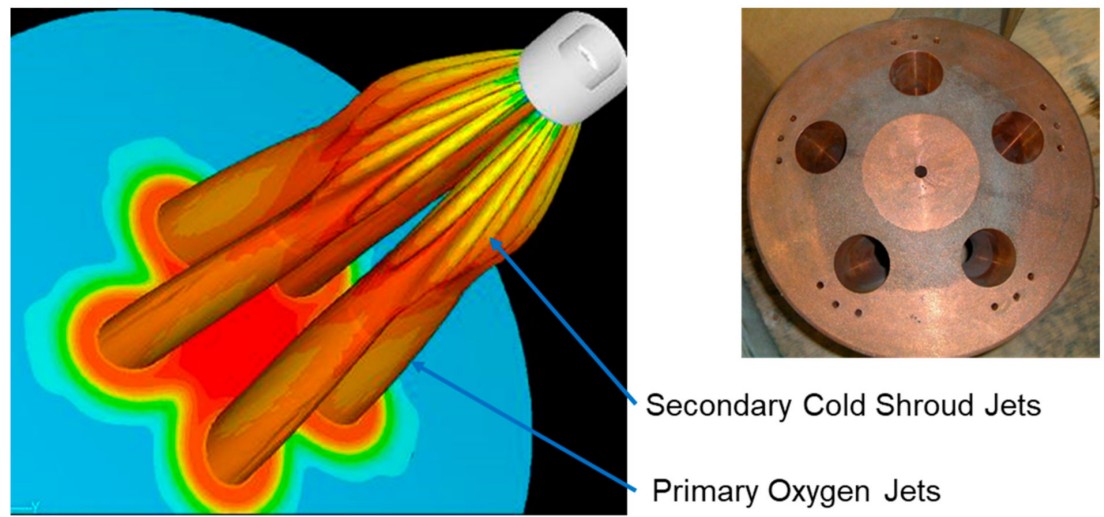

**Secondary Cold Shroud Jets**

**Primary Oxygen Jets**

**Figure 14.** The principle of Cold Shroud © technology. Reprinted with permission from Refs. [66,67], 2022, Tallmantechnologies.

Publications on post-combustion in practical use in the last decades are also related to reliable and safe operation. The authors reported the reduced formation of lance skulls and build-up at the furnace mouth and cone, reducing downtimes and enhancing operational safety. Lances with special designs were applied for this (Valentas et al. [68], Egger et al. [32] Lehner et al. [26], and Gillgrass et al. [69]).

An alternative approach for the increase of scrap by post-combustion technique is the Jet Process, which comprises a hot blast lance in combination with oxygen bottom blowing. Primetals owns the process, and its scheme is shown in Figure 15. Coal and lime can be injected via the bottom tuyere. The blast, oxygen-enriched air is preheated regeneratively in a pebble heater. During blowing, the heat is released to the blast, and during converter tapping and charging of the next heat, the pebble heater is reheated with hot flue gas generated by the combustion of natural gas with air. The bottom oxygen oxidizes the dissolved carbon in the hot metal and the coal to CO. An extensive post-combustion of CO to $CO_2$ can be achieved through the injection of the hot blast. By these measures, coal and hot blast injection, the hot metal charge ratio should be decreased to 50% of the total metallic charge. The input of solid ferrous metals, i.e., scrap and sponge iron, can be increased accordingly. $CO_2$ emission of 1000 kg $CO_2$/t crude steel is claimed for the Jet Process compared to 1600 kg $CO_2$/ton crude steel for the conventional LD-process [70,71].

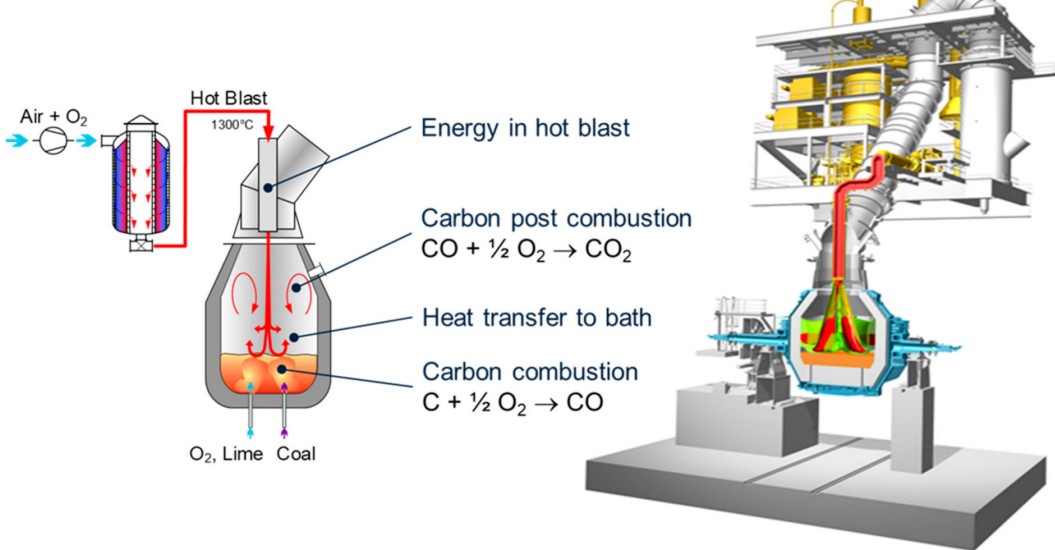

**Figure 15.** Process scheme of the Jet Process. Reprinted with permission from Ref. [71], 2022, Primetals.

In a joint venture between Saarstahl AG, Germany, and POSCO, Korea, the process principle of the Jet Process was proven in 150 trials in 2013. For this purpose, a 100-ton LD converter in Pohang works in Korea was equipped with a pebble heater, hot blast lance, and bottom oxygen tuyeres. One of the main findings was that the hot metal ratio could decrease to 76% with hot blast operation. The effect of coal injection was not published [72].

A hybrid solution combining oxygen refining and electric energy supply by arcing is the CONARC process owned by the German plant builder SMS Group GmbH. The name comes from combining CONverter and ARCing. The process concept provides high flexibility in the charge mix of hot metal, scrap, and DRI. The CONARC furnace is equipped with a double vessel system and a swiveling system for an electrode and oxygen lance. In this way, the use of solid iron input, scrap, and DRI can be increased compared to the conventional LD-process. The process runs in two stages (refer to Figure 16): (1) LD converter process to decarburize the liquid hot metal, (2) electric arc furnace process for melting scrap or DRI/HBI and heating the melt to tapping temperature. The shape of the vessel resembles that of an LD converter.

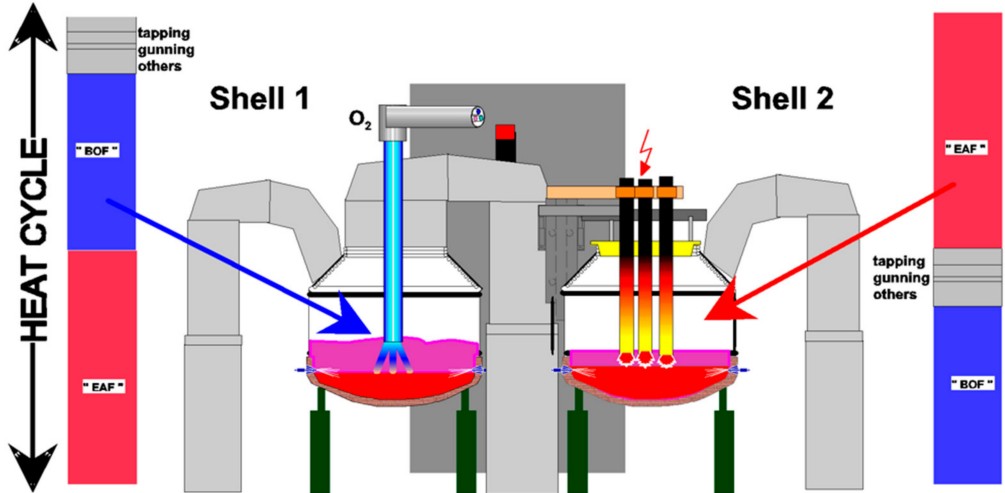

**Figure 16.** Process concept of CONARC. Reprinted with permission from Ref. [73], 2022, SMS Group GmbH.

### 4.2. Use of By-Products

The LD-processes produced 85 to 165 kg, typically 125 kg, slag per ton of crude steel as a by-product [74]. The typical use of LD-steelmaking slags is for base course material, aggregate for asphalt, material for civil engineering works, ground improvement material, and raw material for cement clinker and fertilizer and soil improvement. In some countries, environmental authorities issued new regulations limiting LD and EAF steelmaking slag for road constructions or agriculture. Research and developments have been initiated to find new applications and to prevent landfilling [75].

One approach is the modification of the chemical composition with additives. A research team in Germany and Austria developed a new product for the cement industry by changing the basicity of LD slag and controlling cooling from liquid to solid-state [76].

An alternative is the extraction and separation of iron as oxide or metal. A promising approach is to cool the LD slag slowly, producing big iron oxide-rich phases. The slag is then ground and magnetically separated to recover the iron-rich phase [77]. A current development project funded by the EU aims for a new treatment to produce an iron oxide-enriched fraction and a mineral fraction enriched with calcium and phosphorus oxide. The first can be reused within steel production and the second as an additive in the cement or fertilizer industry [78].

The phosphorus recovery from LD slag for fertilizer production is a further valorization option. A Japanese research group investigated the separation and recovery of phosphorus from steelmaking slag via a selective leaching and chemical precipitation process. Japan is currently importing the same amount of phosphorous with iron ore as imported phosphate ore for agriculture [79].

The carbothermic reduction of LD slag to recover phosphorous has been investigated in lab-scale experiments. The phosphorous compound is reduced from the slag in an inductively heated bed with graphite. The formed gaseous elemental phosphorous is condensed in a scrubber with water and forms phosphoric acid, which is further used for calcium phosphate production. A scale-up of this concept is underway [80].

On average, 1 to 24 kg, typically 13 kg, of dust per ton of crude steel is produced as a further solid by-product in the LD-process [74]. The coarse dust fraction can be recycled internally in the integrated route. It can be charged directly back to the LD converter or sent to the sinter plant, and in the form of briquettes to the blast furnace. The fine fraction has to be treated externally and is partly landfilled. Voest-alpine in Linz is recycling briquetted dust to the converter to enrich the zinc content in the dust. These briquettes are removed from the cycle if the zinc content is sufficiently high and sent to external zinc recovery with the DK process or the Befesa Waelz process.

The DK process can recycle zinc-rich metallurgical residuals (3 wt% zinc on average and up to 10 wt%) to produce a zinc oxide concentrate (65 to 68 wt% zinc) and pig iron. The Waelz process can treat metallurgical residuals with high zinc content (20 wt% and more). The product is Waelz oxide with a zinc content of about 70%. The by-product is Waelz slag, which can be used as road construction material. One disadvantage of the Waelz process is that iron is bound in the slag and cannot be recovered and further utilized, e.g., in steel production as an iron carrier. Alternative technologies that can process metallurgical residues with zinc and produce zinc and iron-rich fractions are the Oxicup, Primus, and Redsmelt$^{TM}$/Rediron$^{TM}$ processes [81]. A new development is the Reco Dust process, which was tested on a pilot scale. It can produce a crude zinc-oxide with about 65 wt% zinc and iron oxide-rich slag, which can be recycled at the sinter plant in integrated steelworks [82]. The processing of high-Zn containing dust is a key topic for LD steel plants as today the use is limited, but the scrap quality with the lowest residual levels is excellent.

### 4.3. Process Automation

Process automation systems have significantly contributed to the highest product output at the lowest cost for LD steel plants' operations worldwide. The process models provide tools to optimize raw material consumption, enhance availability, and meet crude

steel quality specifications. They contribute to reducing emissions and hazards for health and safety. Several metallurgical plant builders offer dynamic process control systems to predict the endpoint carbon content in the crude steel before tapping. Time series of process data such as lance movement and CO and $CO_2$ level in either the off-gas or spectral analysis of the flame in the hood are collected during refining. They are online processed with mathematical models. Typically, these model calculations and measurements with the sublance system are computed to predict the endpoint carbon content.

With the increased computing capacity in recent years, intelligent models for end-point control were developed. More complex physicochemical models are introduced, allowing the dynamic calculation of the refining reaction for carbon, silicon, manganese, and phosphorous. Sub-models compute thermodynamic equilibria in the metal and slag; reaction kinetics between metal, slag, and gas phase; and the dissolution kinetics of scrap and slag-forming additives. A feature of intelligent models is data-driven machine learning combined with big data analytic techniques. These models can predict crude steel composition and temperature as well as the slag composition at the end-blow point.

In the coming years, it can be expected that the capability of intelligent models will be further improved. The development of new measuring and detection systems is ongoing. Robotic systems are already used, but their application will become more widespread. Continuous and wireless measuring systems for temperature and composition are under development and will provide additional online data for the model.

Expert systems monitor the metallurgical process and support the operator's decisions. They provide suggestions based on model calculations and the stored production schemes for different steel grades. The expert systems will benefit from new, more powerful intelligent process models. Models based on artificial intelligence are not yet applied in operational practice. Application in plants that already use sophisticated expert models is rather doubtful, but this technology seems promising for plants at the ignition point of process automation. Digitalization will further enhance the reliable and safe operation of the LD-process in the following decades.

## 5. Future Aspects of LD-Steelmaking

Today, the global steel industry is heading towards a major transition phase. This process has been triggered by the attempt to reduce greenhouse gas (GHG) emissions and thus to avoid the consequences of climate change. The global steel industry is responsible for about 7% of the worldwide GHG emissions [83], and fossil fuels resulting in $CO_2$ are especially in the public focus. For industrialized regions, steel production is often the largest single $CO_2$-emission source [84]. Thus, the de-carbonization of the steel industry is not only a huge challenge, but also a tremendous opportunity for GHG reduction, and $CO_2$ is the main focus.

Most $CO_2$ emissions from steelmaking are caused by the so-called "primary" steel route, which uses iron-ore as a major iron source. Today, this route is dominated by blast furnaces combined with LD-steelmaking, and about 2.2 tons of $CO_2$ are emitted for every ton of crude steel [83].

It is an easy but wrong assumption that EAF steelmaking based on recycled scrap could replace LD-steelmaking on a worldwide scale and lead to lower $CO_2$ emissions. In fact, LD-steelmaking processes already utilize relevant amounts of recycled scrap, and it is well understood that the use of recycled scrap alone cannot cover the world's steel demand. That is true today because, of the 1.9 billion tons of steel production worldwide, 1.4 billion tons is converter steel [85], and this will remain valid according to relevant forecasts [83]. Steelmaking from iron ore will remain crucial, and today the LD-process is the unavoidable core process of high-quality steelmaking. Therefore, it is neither clear nor decided how the future of LD-steelmaking will be, and it is worth analyzing the challenge of carbon neutrality in more depth.

Since international treaties such as the Paris Agreement from 2015 [86] were translated into $CO_2$-pricing schemes on national levels, $CO_2$ avoidance became an active economic

factor in steel production. Especially in the case of strategic investment decisions, $CO_2$ emissions per ton of steel produced have become a key factor within the steel industry. In addition, steel producers worldwide have committed to $CO_2$-reduction targets [83] and will be held accountable by their stakeholders accordingly. All metallurgical production routes are on trial. Like all established processes, the LD-process needs to prove its viability in this new context.

Immediate $CO_2$ reductions are often the first focus, and the LD-process offers essential opportunities for prompt $CO_2$ avoidance. The already beneficial use of scrap within the LD-process can be further enhanced by post-combustion. Digitalization of metallurgical processes enables more effective energy and raw material utilization and thus reduce $CO_2$ emissions. However, even relevant $CO_2$ savings are not enough to reach carbon neutrality. Will LD steelmaking only be a bridging technology, or will it master the challenge of carbon neutrality? This question needs far more detailed considerations. In the following, it will be outlined that the LD-process can play a decisive role in future carbon-neutral steelmaking.

In Figure 17, parts of an in-depth analysis [87] of emissions from the classical production of blast furnaces followed by LD-steelmaking and subsequent hot rolling are presented. The values are compliant with the Life-Cycle-Assessment (LCA) standard ISO 14040/44. While the whole production chain from producing raw materials to rolled coil was included in the original work, the focus in the figure lies naturally on emissions that emerge at the steelmaking site itself and are within the direct control of the steel producer. Those so-called direct emissions generate a global warming potential (GWP) of 1.9 tons $CO_2$-equivalent per ton of hot-rolled coil. Within LCAs, the emissions are attributed to the location where they enter into the atmosphere. In fact, most emissions in an integrated steel mill leave the site via the power plant, as can be concluded from Figure 17.

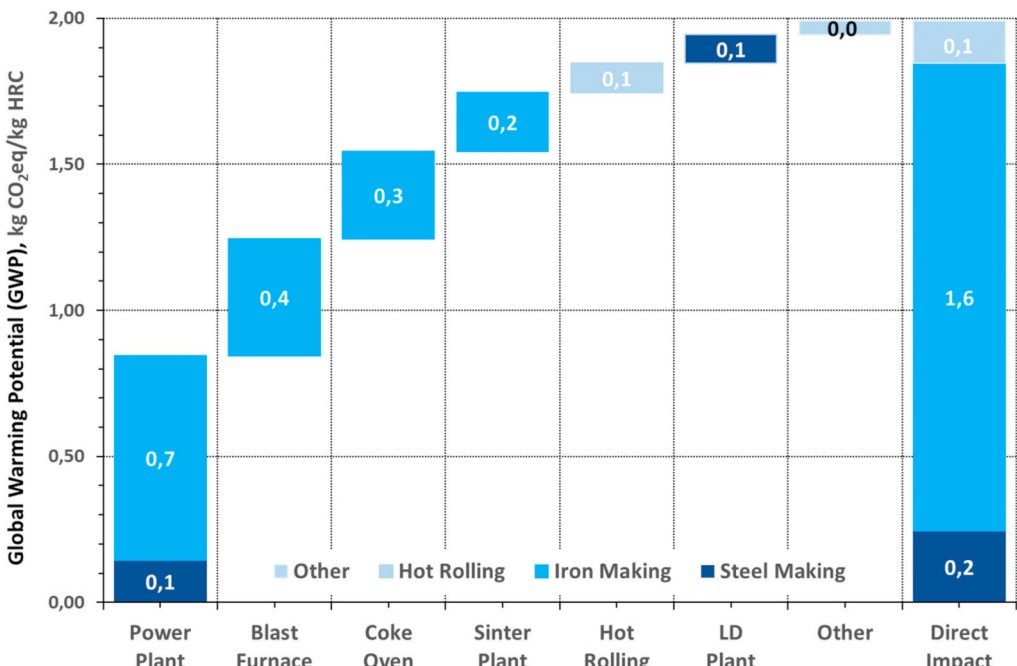

**Figure 17.** Carbon dioxide-equivalent (Global Warming Potential) from contemporary Integrated steelmaking.

The LCA methodology is by far the most common and, at the same time, the most reliable and accepted way of accounting emissions. But the fact that most $CO_2$ emissions leave steel mills via the power plant does not provide an answer on how to avoid these emissions. Only additional process knowledge reveals where the origin of fossil carbon usage lies. Gases, which are used inside the power plant, come inevitably from the metallurgical processes themselves. For this reason, Figure 17 delivers an additional but rough idea of the origin of carbon usage. Emissions are assigned to the process steps: hot metal

production, oxygen steelmaking, and hot-rolling. It becomes apparent from Figure 16 that most $CO_2$ emissions, 1.6 kg of $CO_2$-equivalent per kg of hot-rolled coil (HRC), originate from hot metal production. Oxygen steelmaking, i.e., the LD-process, has a limited share of 0.2 kg, and hot-rolling has an even smaller contribution of only 0.1 kg.

As can be concluded from Figure 17, the first focus of de-carbonization of the steel industry needs to focus on hot metal: contemporary hot metal production via blast furnaces does not only use fossil carbon for heating, reduction, and melting. In fact, the blast furnace process needs relevant excess fossil carbon, leaving the blast furnace as a high calorific gas. In the past, this inability to utilize carbon fully inside the main process was not seen as an inefficiency because the high calorific gas could be transformed into electrical energy that was required on the steel mill side.

Today, new assumptions overwrite the traditional answers. In a "fossil-free" world, it is believed that Green Hydrogen and Green Electricity are available in sufficient amounts. Carbon usage is not forbidden but restricted to biogenic carbon from either circular sources or any other means of re-transforming $CO_2$ back into carbon; this carbon will be limited and expensive. The obvious first task in the attempt to establish carbon-neutral steel production, respecting these conditions, is a fossil-free reduction and melting step. Such an undertaking is hardly possible with today's blast furnace technology, and the consistent conclusion today is a replacement of blast furnaces by direct reduction technology [88–90]. This understanding includes a second inherent conclusion: there is no apparent need to abandon the LD-process. In fact, the direct $CO_2$ emissions of LD-refining are in the same order as EAF melting. Both technologies rely on a certain amount of metallurgical carbon. Carbon will remain the main component for steel strength properties and will still promote certain refining reactions, so carbon cannot be eliminated entirely from any steel production route. "Zero Carbon" expresses the aim to avoid any increase in GHG within the atmosphere. It should not be misunderstood as "No-Carbon". Such necessary carbon ought to be provided from $CO_2$-neutral sources in the future. The availability of climate-neutral hydrogen vs. climate-neutral carbon can be a strong future influence on how metallurgical processes will develop.

The forward-looking concept of steelmaking routes that combine LD converters with direct reduction (DR) plants has already been described in the literature [91–94]. The flow process chart introduces an intermediate melting step for directly reduced iron, which delivers an iron melt with some carbon content; see Figure 18. Such pre-melt is sometimes called "Electric-Hot-Metal" as it is similar to classical hot metal in chemical composition. It is evident that this pre-melt can be transported by existing torpedo cars. It even can be mixed with classical hot metal from the blast furnace while—within a transition phase—DR plants and blast furnaces work side by side on existing integrated mills. Much like today, electric and classical hot metal is eventually processed in the LD converter. Electric-Hot-Metal will feed the LD converters of the future.

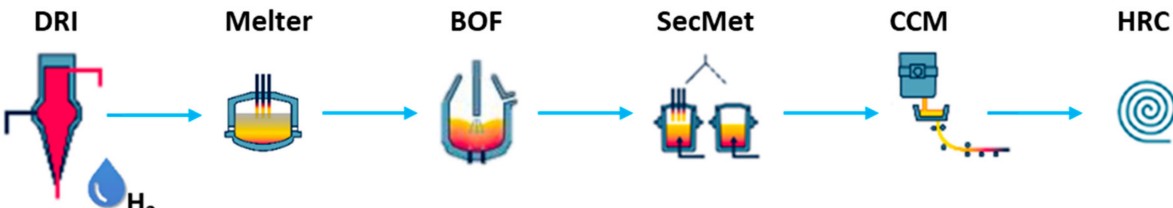

**Figure 18.** Possible LD concept for future steelmaking. Process Route combining DR-Plant, electric melting, and LD converter.

Like the already realized combination of DR Plants with classical EAF, the combination of DR plants with LD converters [94] can reach climate neutrality on the basis of green energy. The use of LD converters in the special configuration, as shown in Figure 17, offers some interesting strengths, both technological and economical:

(a)    It is immediately evident that the LD concept allows continued operation of existing melt shops and subsequent continuous casting and rolling. Fewer investment costs for a transition of existing sites are one of the important consequences. All investment into existing integrated sites can concentrate on Electric-Hot-Metal production.

(b)    Continuous casting of slabs is known to deliver the best available surface qualities, and it is still unclear if mini-mill concepts with EAFs can close this gap in the far future. Therefore, keeping existing steel mills and integrated sites is superior to a new investment in a mini-mill concept.

(c)    When it comes to chemical composition, the LD converter can play to its strengths within the new configuration: lowest phosphorous and lowest nitrogen, especially combinations of lowest carbon and lowest nitrogen. The LD converter will continue to enable the most sophisticated chemical compositions for the final steel product. Metallurgical experts are well aware that the ability to "repair" chemical composition within the secondary metallurgy is limited.

(d)    The melting step is stripped from unnecessary metallurgical work. Robust equipment that follows the idea of submerged-arc furnaces run under a reducing atmosphere and offers some unique opportunities in raw materials that are not part of EAF steelmaking. Certainly most important is that there is no need to use DR grade pellets as feedstock inside the DR plant as there is no need for a foaming slag within the melting aggregate. Availability of DR grade pellets is already a topic today and is likely to get worse in the future. Besides directly reduced iron, iron-bearing recycling material can also be introduced, and even scrap can be added. The subsequent slag from this process step can be handed over to the cement industry, as happens today with blast furnace slag.

(e)    Finally, the LD-process will form a completely new production chain in combination with new upstream processes. This new route offers opportunities to deal with fluctuations in energy supply much better than the traditional combination with blast furnaces did. While this aspect is mainly irrelevant today, it might be decisive in a "fossil-free" future.

## 6. Summary

According to the historical records, on 27 November 2022, the LD-process will celebrate its 70th anniversary after the first heat was blown in the first LD-plant of the world in Linz (the name LD was formed from the 1st letters of the two sites with the first industrial scale plants: Linz and Donawitz, both in Austria). Since its start with 30-ton converter sizes, the evolution progressed quickly, and the size of the vessel increased and, with the size, the productivity of the plants. The biggest converters today have >400 tons/heat of tap weight; a size of 330–350 tons/heat seems to be the best fit for the process chain in modern plants. The first generation of LD-plants were designed for a yearly capacity of 450,000 tons/year. The biggest plants in the world today produce around 6,000,000 tons/year; in Asia, giants plants are producing >12,000,000 tons/year. Already in the late 1970s, the share of the LD-process on crude steel production worldwide had reached >60%. In the peaking crude steel production, which has more than doubled since the year 2000, the LD-process has maintained its leading position and has increased its market share to >70% today.

Numerous technical developments have improved the productivity, reliability, and sustainability of the process to state-of-the-art technology, which is described in the paper in detail. Different process operation philosophies have been established in the regions of the world based on the local raw material availability and product quality requirements. There are still substantial technological advances left that will reduce the impact of the LD-process on nature, society, the value chain, and the workforce. Major work is carried out for scrap rate increase and reduction and reuse of the by-products slag and dust, as well as process digitalization and automation. Upon the automatic blowing process today, auto-charging and auto tapping are also already applied in some of the plants.

The biggest challenge to overcome over the coming decades is the reduction of the $CO_2$ footprint of ironmaking and steelmaking to contribute to the climate change targets of the world society. It is a common understanding that the properties of the material steel are directly linked to its carbon content. Only a few very special grades (stainless steel, ULC) can be considered as carbon-free. The oxygen steelmaking process is also addicted to carbon as the heat generation of the process relies on carbon (and other elements) oxidation. Additionally, the refining of virgin iron ore phosphorous with liquid slag requires the use of oxygen. Without any carbon, the iron loss would be too high and make the technology uneconomical.

A new idea to produce "Electric-Hot-Metal" from hydrogen-reduced iron, which is molten in an electrical smelter, will open the opportunity to provide liquid hot metal to existing LD-process facilities and utilize the already existing infrastructure of the integrated mills with the installed casting machine and rolling equipment. This strategy, amongst others, incorporates considerably fewer new product certification/homologation efforts, which are a common procedure in industrial supply chains to avoid negative impacts on the downstream industries by technology changes upstream.

We congratulate the jubilarian on its 70th birthday and look forward to continuing a successful, exciting life together.

**Author Contributions:** Conceptualization, J.C.; methodology, J.C., M.W.E. and F.A.; investigation, J.C., F.A., M.W.E., H.H. and J.S.; data curation, J.C., F.A. and M.W.E.; writing—original draft preparation, J.C., F.A., M.W.E., H.H. and J.S.; writing—review and editing, J.C., F.A., M.W.E. and J.S.; visualization, J.C., F.A. and M.W.E.; supervision, J.C. and J.S. All authors have read and agreed to the published version of the manuscript.

**Funding:** This research received no external funding.

**Institutional Review Board Statement:** Not applicable.

**Informed Consent Statement:** Not applicable.

**Data Availability Statement:** Data sharing is not applicable.

**Conflicts of Interest:** The authors declare no conflict of interest.

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
