# Peer review of "70 Years of LD-Steelmaking—Quo Vadis?"

_metals, doi:10.3390/met12060912_

Round 1

Reviewer 1 Report

The authors review the story of the LD-process, introduce the recent technology developments and give an outlook on the future role of the process in the steel industry. It is a good review manuscript. I think that the history and development of the LD process were published, It is not necessary to introduce in detail. Please consider deleting some unimportant discussion.

Author Response

Refer to attachment

Reviewer 2 Report

This article summarizes the 70-year history of the development of LD-steelmaking, introduces the background of the emergence of the converter in detail, and also introduces the progress of the supporting facilities and technology of the converter, which provides a wealth of knowledge for related researchers and workers.

There are some issues in the manuscript that need to be discussed and it should be MINOR REVISION. This choice is made mostly on the basis of the following reasons.

  1. In the second section, some content may be appropriately added, such as the key innovative work and contributions made by relevant researchers in various countries in the LD-process.
  2. How does the author view the advantages and disadvantages of EAF steelmaking and converter steelmaking? What is the author's opinion as the most important limiting factor in EAF steelmaking at present? Which process will the production method of steel in the future mainly rely on?
  3. There may be some details in the text that need to be corrected. For example, the text at the top of Figure 1 lacks a punctuation mark, and the two pictures in Figure 6 may have overlapping parts. It is recommended that the author read the full text to correct similar details.
  4. In section 15, the author mentions the current data on the annual output of converter tapping, and suggests that the author supplement the data source.

Author Response

Refer to attached file

Reviewer 3 Report

This is a good review into the history of LD process. The weight is clearly in the past. The title points to future progress, but this part is short and remains on a qualitative level ending up positive future outlooks for LD-type processes. Cconsideration on eventual combinations of BOF and EAF processes is promising but already existing technologies/processes like Conarc were not mentioned. One bigger "technical" error concerns the references. In the text they should be in the order 1,2,3... Here they are in random order (the 1st one is 6) and numerous ref numbers are erroneous i.e. the citation cannot be to the given number. I don´t list the most evident cases, but you should check the all and  put in correct order.

The list of smaller comments, errata or proposed revisions are here.

P. 1 How it started line 5 double in in. In this part reference missing. P. 2 line 13: Donwitz -> Donawitz

P. 4 line 10:...denied _> rose above (reviewer´s proposal). Figure 3. Development of the world steel production and the share...Only the share is shown, but the steel production is missing! Below Fig. 3 CRNM ,,,it used to be CRM

P. 5 on the top. You claim magnesia lining was established around the globe. You maybe simplify the progress. Combination lining with doloma, doloma-magnesia, magnesia optimizing the economic and productional aspects has been the main trend over the decades.

P. 6 Figure 4: The red line remains unclear as well as the scale on y-axis on the right side. Reference? 2. State-of-the-art in Technology; the chapter is too short. Figure 5 earns little more comments.

P. 8 Figure 7: line 3: 72-95%; in Fig. 800-1060kg/t liquid. Are these figures comparable? Some dots have marks VASL etc but mostly not. Explain or complete.

P. 9 lin3 3: Figure 7 -> 8

P. 11 9.3 Doble-> Double

P. 12 above Fig. 13 you discuss about 1980-90 but the ref. is from 1959?

P. 15 line 5-6; the figures 1000/1600 depend on the system boundaries; what is included/excluded. By defining the boundaries, the credibility would increase.

P. 16 line 11: 1 to 24 kg, Can it be only 1 kg?

P. 20 Summary line 7 Ltons/heat; is L a typing error? line 12-13: In the following the peaking crude steel production...Comment: what does "peaking" mean here? As you well know, the true peaking of steel production is far in the future.

Author Response

Refer to attached file
